**SOFTWARE**                                                                         **Open Access**

# FAN-C: a feature-rich framework for the analysis and visualisation of chromosome conformation capture data

Kai Kruse[1], Clemens B. Hug[1] and Juan M. Vaquerizas[1,2]*

* Correspondence: j.vaquerizas@lms.
mrc.ac.uk; jmv@mpi-muenster.mpg.
de
[1]Max Planck Institute for Molecular
Biomedicine, Roentgenstrasse 20,
48149 Muenster, Germany
[2]MRC London Institute of Medical
Sciences, Institute of Clinical
Sciences, Faculty of Medicine,
Imperial College London, Du Cane
Road, London W12 0NN, UK

## Abstract

Chromosome conformation capture data, particularly from high-throughput approaches such as Hi-C, are typically very complex to analyse. Existing analysis tools are often single-purpose, or limited in compatibility to a small number of data formats, frequently making Hi-C analyses tedious and time-consuming. Here, we present FAN-C, an easy-to-use command-line tool and powerful Python API with a broad feature set covering matrix generation, analysis, and visualisation for C-like data (https://github.com/vaquerizaslab/fanc). Due to its compatibility with the most prevalent Hi-C storage formats, FAN-C can be used in combination with a large number of existing analysis tools, thus greatly simplifying Hi-C matrix analysis.

**Keywords:** Chromosome conformation capture, Hi-C, Hi-C analysis, Topologically associating domains (TAD), Chromosomal compartments, Chromatin loops, Hi-C visualisation

## Background

The development over the last decade of high-throughput techniques to study the three-dimensional organisation of the genome [1–3] in the nucleus has fuelled the characterisation of chromatin conformation in a wide variety of biological systems. These range from the organisation of the bacterial nucleoid [4], to the in vitro characterisation of the molecular mechanisms that govern chromatin organisation in eukaryotes [5–10], reviewed in [11], how this organisation is dynamically regulated during cell cycle [12, 13], development and differentiation [14–19], reviewed in [20], and how it is affected in disease [21–23], reviewed in [24].

Given the fundamental role that the correct organisation of chromatin in the nucleus plays for proper cell physiology, there is a growing need to integrate chromatin contact data in current studies examining different aspects of gene and genome regulation. Different techniques have been developed to study chromatin conformation at the single cell or population level, in situ Hi-C being the primary method of choice for analysing chromatin conformation in cell populations [25], reviewed in [26] (Fig. 1a, left).

**Fig. 1** Overview of FAN-C functionality. **a** Overview of Hi-C from an experimental (left) and computational (right) perspective. RE: restriction enzyme. **b** Matrix generation features. **c** Hi-C matrix analysis features. **d** Hi-C visualisation features. **e** Helper tools

The large amounts of Hi-C data and increasingly specialised research questions have led to the development of diverse Hi-C analysis tools. Typically, these fall into one, rarely multiple of the following categories: Hi-C matrix generation, feature analysis, and visualisation [27–29] (Fig. 1a, right). Hi-C matrix generation tools convert raw sequencing reads from a Hi-C experiment into a normalised matrix of interaction strengths between pairs of genomic regions, accounting for biases and false-positive interactions in the process. Feature analysis tools act on the Hi-C matrix to derive measures, models, and statistics that answer specific biological questions, such as the identification of topologically associating domains [30–32] and chromatin loops [25, 33], the 3D modelling of the chromatin fibre [34, 35], or the identification of differential contacts between samples. Visualisation tools then enable the static display, and sometimes interactive exploration of the Hi-C matrix, often together with associated genomic data derived using other methods, such as ChIP-seq [29, 36].

The complexity of handling Hi-C data, owed in part to the vast amounts of data produced, prompted the development of several dedicated Hi-C storage formats in the form of compressed binary [37] or hierarchical files [37], or as text file specifications. The combination of specialised tools and available Hi-C formats results in a fragmentation of Hi-C analysis methods, which in turn causes a significant overhead for researchers analysing Hi-C data (Table 1).

Here, we present FAN-C, a <u>F</u>ramework for the <u>AN</u>alysis of <u>C</u>hromatin Conformation Capture data, an easy-to-use command-line tool and powerful Python API with a broad feature set covering matrix generation, analysis, and visualisation (Fig. 1). FAN-C uses a custom hierarchical storage format optimised for fast matrix access and common Hi-C matrix transformations. In addition, it is natively compatible and inter-convertible with the widespread Cooler [38] and Juicer [37] Hi-C file formats and can import a large variety of different text-based matrix inputs, such as those generated by HiC-Pro [42]

**Table 1** Feature comparison of different Hi-C analysis tools. Tools included in the comparison are Cooler [38]/HiGlass [39], Juicer [37]/Juicebox [40], HOMER [41], HiC-Pro [42], HiC-bench [43], TADbit [44], HiFive [45], HicDat [46], HiC-inspector [47], HiCUP, HiCInspector [47], HiCUP, HiCExplorer [48, 49], and HiCeekR [50]. 1: Only for interactive plotting; 2: Support for Juicer and Cooler multi-resolution files, but no native support; 3: Cooler ecosystem includes pairtools, cooler, cooltools, HiGlass, and distiller; 4: In conjunction with Juicebox; 5: Provides instructions for mapping, but no dedicated command; 6: Visualisation through Treeview; 7: With export for Fit-Hi-C; 8: Through compatibility with HiCPlotter; 9: Via HiCNorm; 10: Fit-Hi-C, C-loops, and targeted virtual 5C (in-house); 11: Only pre-processing; 12: For interactive visualisation; 13: SAM/BAM visualisation through SeqMonk; 14: via pyGenomeTracks; 15: Only when previously marked in BAM file; 16: via spacewalk; 17: no dedicated function, but possible via API; 18: Via Galaxy; 19: Includes hierarchical clustering of TADs; 20: Personal communication by developers, not currently documented; 21: insulation and compartment scores; 22: via TADkit

| | FAN-C | Cooler[3] | Juicer | HOMER | HiC-Pro | HiC-bench | TADbit | HiFive | HicDat | HiC-inspector | HiCUP | HiCExplorer | HiCeekR |
|---|---|---|---|---|---|---|---|---|---|---|---|---|---|
| **User interfaces** | | | | | | | | | | | | | |
| Command line | × | × | × | × | × | × | × | × | | × | × | × | |
| Programmatic access (API) | × | × | × | | | | × | × | × | | | | |
| Graphical user interface (GUI) | 1 | 1 | 4 | | | | 1 | | 11 | 12 | | 18 | × |
| **Supported formats** | | | | | | | | | | | | | |
| **Directly compatible** | | | | | | | | | | | | | |
| Juicer | × | | × | | | 20 | | | | | | | |
| Cooler | × | × | | | | 20 | | | | | | × | |
| Custom/Native | × | × | × | × | × | × | × | × | × | × | × | × | × |
| **Import** | | | | | | | | | | | | | |
| Juicer | × | × | × | | × | | | | | | | × | |
| Cooler | × | × | | | × | | × | | | | | × | |
| TXT file | × | × | × | × | × | × | × | × | | × | | × | |
| FASTQ | × | × | × | 5 | × | × | × | | 5 | × | × | × | |
| SAM/BAM | × | × | × | × | × | × | × | × | × | × | × | × | |
| hiclib | × | × | | | | | | | | | | | |
| **Export** | | | | | | | | | | | | | |
| Juicer | × | | × | × | × | 20 | × | | | | × | | |
| Cooler | × | × | | | × | 20 | × | | | | | × | |

**Table 1** Feature comparison of different Hi-C analysis tools. Tools included in the comparison are Cooler [38]/HiGlass [39], Juicer [37]/Juicebox [40], HOMER [41], HiC-Pro [42], HiC-bench [43], TADbit [44], HiFive [45], HicDat [46], HiC-inspector [47], HiCUP, HiCInspector [47], HiCUP, HiCExplorer [48, 49], and HiCeekR [50]. 1: Only for interactive plotting; 2: Support for Juicer and Cooler multi-resolution files, but no native support; 3: Cooler ecosystem includes pairtools, cooler, cooltools, HiGlass, and distiller; 4: In conjunction with Juicebox; 5: Provides instructions for mapping, but no dedicated command; 6: Visualisation through Treeview; 7: With export for Fit-Hi-C; 8: Through compatibility with HiCPlotter; 9: Via HiCNorm; 10: Fit-Hi-C, C-loops, and targeted virtual 5C (in-house); 11: Only pre-processing; 12: For interactive visualisation; 13: SAM/BAM visualisation through SeqMonk; 14: via pyGenomeTracks; 15: Only when previously marked in BAM file; 16: via spacewalk; 17: no dedicated function, but possible via API; 18: Via Galaxy; 19: Includes hierarchical clustering of TADs; 20: Personal communication by developers, not currently documented; 21: insulation and compartment scores; 22: via TADkit *(Continued)*

| | FAN-C | Cooler[3] | Juicer | HOMER | HiC-Pro | HiC-bench | TADbit | HiFive | HicDat | HiC-inspector | HiCUP | HiCExplorer | HiCeekR |
|---|---|---|---|---|---|---|---|---|---|---|---|---|---|
| GInteractions | | | | | | | | | | | | × | |
| TXT file | × | × | × | × | × | × | × | | | × | × | × | × |
| **Matrix generation** | | | | | | | | | | | | | |
| **FASTQ mapping** | | | | | | | | | | | | | |
| Simple mapping | × | × | × | 5 | × | × | × | | 5 | × | × | 5 | |
| Iterative mapping | × | | | | | | × | | | | | | |
| Ligation junction split | × | | × | × | × | | × | | | | × | | |
| **Read/region pair filtering** | | | | | | | | | | | | | |
| Mapping Quality | × | × | × | | × | × | × | | | | × | × | |
| Multi-mapping reads | × | × | × | | × | × | × | | | | × | × | |
| Restriction site distance | × | × | × | × | × | × | × | × | | × | × | × | × |
| Ligation errors | × | × | × | | × | | × | × | × | | × | × | × |
| Self-ligations | × | × | × | × | × | × | × | × | × | | × | × | × |
| PCR duplicates | × | × | × | × | × | × | × | × | | | × | × | 15 |
| Unusual read density | | × | | × | | | × | | | | | | |
| Quality statistics | × | × | × | × | × | × | × | × | | × | × | × | |
| **Hi-C processing** | | | | | | | | | | | | | |
| Fragment-level Hi-C | × | × | × | × | × | × | × | × | × | | × | × | |
| Equi-distant bins | × | × | × | × | × | × | × | × | × | × | | × | × |

**Table 1** Feature comparison of different Hi-C analysis tools. Tools included in the comparison are Cooler [38]/HiGlass [39], Juicer [37]/Juicebox [40], HOMER [41], HiC-Pro [42], HiC-bench [43], TADbit [44], HiFive [45], HicDat [46], HiC-inspector [47], HiCUP, HiCExplorer [48, 49], and HiCeekR [50]. 1: Only for interactive plotting; 2: Support for Juicer and Cooler multi-resolution files, but no native support; 3: Cooler ecosystem includes pairtools, cooler, cooltools, HiGlass, and distiller; 4: In conjunction with Juicebox; 5: Provides instructions for mapping, but no dedicated command; 6: Visualisation through Treeview; 7: With export for Fit-Hi-C, 8: Through compatibility with HiCPlotter; 9: Via HiCNorm; 10: Fit-Hi-C, C-loops, and targeted virtual 5C (in-house); 11: Only pre-processing; 12: For interactive visualisation; 13: SAM/BAM visualisation through SeqMonk; 14: via pyGenomeTracks; 15: Only when previously marked in BAM file; 16: via spacewalk; 17: no dedicated function, but possible via API; 18: Via Galaxy; 19: Includes hierarchical clustering of TADs; 20: Personal communication by developers, not currently documented; 21: insulation and compartment scores; 22: via TADkit *(Continued)*

| | FAN-C | Cooler[3] | Juicer | HOMER | HiC-Pro | HiC-bench | TADbit | HiFive | HicDat | HiC-inspector | HiCUP | HiCExplorer | HiCeekR |
|---|---|---|---|---|---|---|---|---|---|---|---|---|---|
| Multi-resolution Hi-C | 2 | x | x | | | 20 | | x | | | | x | |
| Matrix balancing | x | x | x | x | x | x | x | x | x | | | x | x |
| Probabilistic normalisation | | | | | | | | x | x | | | | |
| Matrix merge | x | x | x | | | x | x | | x | | | x | |
| Allele-specific matrices | | x | x | | x | | | | | | | | |
| Mixed restriction cut sites | x | x | x | | x | | | | | | | x | |
| **Hi-C filtering** | | | | | | | | | | | | | |
| Minimum coverage | x | x | | | x | | x | x | | | | | |
| Diagonal | x | x | | | x | x | | | | | | | |
| **Matrix analysis** | | | | | | | | | | | | | |
| **Comparisons** | | | | | | | | | | | | | |
| PCA (sample comparison) | x | 17 | | | | 21 | | | | | | | |
| Matrix fold-change | x | x | x | | | 20 | | | x | | | x | |
| Matrix difference | x | 17 | x | | | 20 | | | | | | x | |
| Score/feature comparisons | x | 17 | x | x | | x | x | | | | | x | |
| Correlations | x | 17 | x | x | | x | x | | x | | | x | |
| **Domains** | | | | | | | | | | | | | |
| Insulation score | x | x | | | | x | x | | | | | x | |
| Directionality index | x | x | | | | x | | | | | | | x |

**Table 1** Feature comparison of different Hi-C analysis tools. Tools included in the comparison are Cooler [38]/HiGlass [39], Juicer [37]/Juicebox [40], HOMER [41], HiC-Pro [42], HiC-bench [43], TADbit [44], HiFive [45], HicDat [46], HiCUP, HiCInspector [47], HICUP, HiCExplorer [48, 49], and HiCeekR [50]. 1: Only for interactive plotting; 2: Support for Juicer and Cooler multi-resolution files, but no native support; 3: Cooler ecosystem includes pairtools, cooler, cooltools, HiGlass, and distiller; 4: In conjunction with Juicebox; 5: Provides instructions for mapping, but no dedicated command; 6: Visualisation through Treeview; 7: With export for Fit-Hi-C; 8: Through compatibility with HiCPlotter; 9: Via HiCNorm; 10: Fit-Hi-C, C-loops, and targeted virtual 5C (in-house); 11: Only pre-processing; 12: For interactive visualisation; 13: SAM/BAM visualisation through SeqMonk; 14: via pyGenomeTracks; 15: Only when previously marked in BAM file; 16: via spacewalk; 17: no dedicated function, but possible via API; 18: Via Galaxy; 19: Includes hierarchical clustering of TADs; 20: Personal communication by developers, not currently documented; 21: insulation and compartment scores; 22: via TADkit (*Continued*)

| | FAN-C | Cooler[3] | Juicer | HOMER | HiC-Pro | HiC-bench | TADbit | HiFive | HicDat | HiC-inspector | HiCUP | HiCExplorer | HiCeekR |
|---|---|---|---|---|---|---|---|---|---|---|---|---|---|
| Arrowhead | | | x | | | | | | | | | | |
| TAD calling | x | | x | x | | x | x | | x | x | | 19 | x |
| **Loops** | | | | | | | | | | | | | |
| HICCUPS | x | x | x | | | | | | | | | | |
| Other | | | | x | 7 | 10 | | | | | | | |
| **Common Hi-C analyses** | | | | | | | | | | | | | |
| Expected values | x | x | x | x | | x | x | x | x | x | | x | |
| AB compartments | x | x | x | x | | 20 | x | | x | | | x | |
| Aggregate Hi-C matrices | x | x | x | | | | | | | | | x | |
| 3D modelling | | | 16 | | | | x | | | | | | |
| **Other** | | | | | | | | | | | | | |
| Compaction | | | | x | | | | | | | | | |
| **Visualisation** | | | | | | | | | | | | | |
| Hi-C matrix | x | x | x | 6 | 8 | 8 | 22 | x | x | x | 13 | x | x |
| Triangular Hi-C matrix | x | x | | 6 | 8 | 8 | 22 | x | x | | | x | |
| Other genomic tracks | x | x | x | | 8 | 8 | 22 | | | | | 14 | x |
| Genes | x | x | x | | 8 | 8 | 22 | | | | | | |
| Virtual 4C | x | x | x | | | 8 | | | | | | | |

and the 4D Nucleome project [51]. FAN-C includes a fully automated FASTQ-to-matrix pipeline, which can be adapted to accommodate the complexities and individual requirements of each specific Hi-C analysis, such as different species or analysis parameters. FAN-C also allows running each pipeline step individually, each with numerous customisation options. In addition, due to its broad file format support, FAN-C has the potential to integrate seamlessly with other tools, thereby significantly simplifying existing Hi-C analysis pipelines.

## Results

### Hi-C matrix generation: from raw sequencing output to chromatin contacts

The first component of the FAN-C analysis framework consists of tools for matrix generation (Fig. 1b). This encompasses the mapping of sequencing reads to a reference genome, assignment of mapped reads to restriction fragments and the formation of interacting fragment-pairs, assembly of a fragment-level Hi-C matrix, and binning, as well as normalising that matrix at different resolutions. At each step, false-positive contacts need to be carefully filtered out in order to prevent matrix artefacts.

The primary tool for matrix generation in FAN-C is a fully automated pipeline, executable by a single command: 'fanc auto'. It accepts a variety of automatically recognised input formats, including (i) unmapped reads in paired-end, optionally gzipped FASTQ files; (ii) mapped reads from SAM or BAM files; and (iii) pre-processed read pairs or genomic contacts from other Hi-C pipelines in the form of text files (Fig. 2a–c). We demonstrate the use of 'fanc auto' and its output files on a high-resolution HUVEC Hi-C dataset [25]. FASTQ files are mapped independently to a reference genome using either Bowtie2 or BWA mem—the choice of mapper is detected automatically from the genome index specified. To boost mapping efficiency, FAN-C can automatically detect and split reads at Hi-C ligation junctions, which are created by the cutting and re-ligation of restriction sites. Further improvements to mapping efficiency can be achieved by enabling iterative mapping [54], where unaligned reads are truncated by a small number of base pairs and then attempted to align again (Fig. 2a).

Mapped reads are then paired and assigned to restriction fragments (Fig. 2b). These are computed automatically using the restriction enzyme name and genome FASTA files or can alternatively be supplied via a custom restriction map. Read pairs are filtered for common biases, including, among others, mapping quality, PCR duplicates, different types of ligation errors (Fig. 2d), and unexpected insert sizes (Fig. 2e). The filtering is highly customisable with a large selection of available filters. More targeted, library-specific filters, such as a filter removing contaminating reads from a different organism, can be added as necessary. Advanced users even have the option to define custom filters using the Python API (Fig. 2b). Diagnostic plots with filtering statistics are generated automatically and are useful to inform the user about potential issues regarding the quality of the Hi-C library or the set of parameters chosen for filtering (Fig. 2f).

Valid pairs, i.e. those that have passed the filtering steps above, are assembled into a fragment-level Hi-C matrix, which in turn is binned at various, customizable resolutions. Each binned matrix then undergoes a second round of filtering at the matrix level, including filters for low coverage of matrix bins (Fig. 2g), and is finally corrected

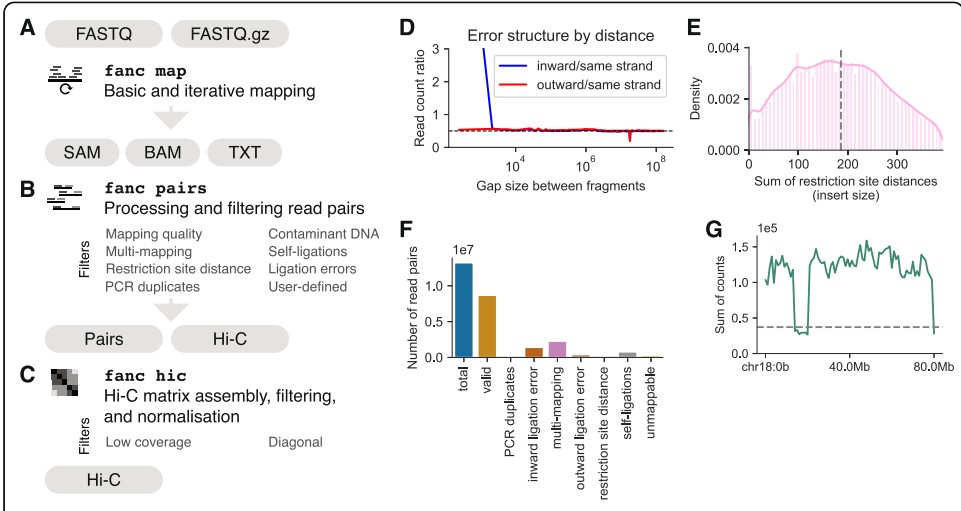

**Fig. 2** FAN-C matrix generation. **a-c** Schematic overview of the matrix generation pipeline. **a** Mapping features. **b** Processing and filtering of Hi-C read pairs. **c** Assembly, filtering and normalisation of the Hi-C matrix from valid read pairs. **d–f** FAN-C statistics plots using data from HUVEC Hi-C [25]. **d** Ligation error plot as in [52, 53]. Dashed line indicates expected values. **e** Density plot of the sum of restriction site distances (insert size) measured from the mapping location of a read to the nearest restriction site. Dashed line indicates median insert size. **f** Summary statistics plot showing the read pairs removed by various filters. **g** Coverage plot of a Hi-C matrix binned at 1 kb resolution. Dashed line indicates the chosen coverage cutoff at 25% median coverage

for experimental and computational biases using Knight-Ruiz matrix balancing [55] or, optionally, iterative correction [54] (Fig. 2c). Importantly, matrix rows and columns whose contact frequencies sum up to zero are explicitly ignored (or in some cases optionally imputed) by all FAN-C analysis methods. This avoids downstream analysis artefacts from falsely treating corresponding regions as if they had a complete lack of contacts, e.g. regions with poor mappability.

One of the key features of FAN-C is the ability to run each pipeline step independently, using dedicated commands. This enables the user to evaluate various parameter settings and to perform parameter sweeps to test the robustness and ensure consistency of their analyses. Importantly, parameter changes can be made after the initial matrix generation, once bias statistics are available and a binned matrix can be investigated, without having to re-run the most time-consuming steps of Hi-C matrix assembly.

While FAN-C was built for utility and compatibility, and not explicitly for performance, it is entirely possible to process and analyse high-resolution Hi-C datasets. Many FAN-C matrix generation steps can be parallelised, and the final cumulative processing time will depend on the resources the user is able to provide. As an example, we processed the GM12878 human B-lymphocyte cell-line dataset [25], containing roughly 5 billion reads. Table 2 lists the approximate processing time on a Linux machine with an AMD Opteron Processor 6376, 2300 MHz, expressed as minutes per 100 million reads, normalised to the processing time on a single thread. Processing times for species with smaller genomes, such as Drosophila melanogaster, can be significantly lower. Once the final Hi-C matrix is obtained, FAN-C can retrieve up to 150 million contacts/pixels per minute from a FAN-C object.

In order to maximise inter-compatibility with existing pre-processing, analysis, and visualisation pipelines, FAN-C includes several conversion tools. Valid pairs can be

**Table 2** Approximate runtimes of the FAN-C matrix generation pipeline. Data from the GM12878 B-lymphocyte dataset [25]. Runtimes are normalised to a single thread, and expressed as minutes per 100 million read pairs. All processing performed on a Linux SGE cluster with AMD Opteron Processor 6376

|  | Minutes/100 M read pairs (single thread) |
| --- | --- |
| **Rao et al. [25] GM12878** |  |
| BWA mem, ligation junction split | 4061.95 |
| Loading + mappability, quality, and uniqueness filters | 264.03 |
| PCR duplicate, RE distance, ligation error filters | 224.15 |
| Fragment-level assembly | 55.72 |
| Merge | 91.09 |
| **Binning, low coverage filter, ICE correction, expected value calculation** |  |
| 1 Mb bins | 167.68 |
| 25 kb bins | 822.15 |
| 5 kb bins | 1455.1 |

converted to Juicer's Hi-C format using 'fanc to-juicer'. Similarly, binned FAN-C matrices can be exported to multi-resolution Cooler files using 'fanc to-cooler', which are then compatible with cooltools [56] and HiGlass [39] for visualisation.

### Matrix analysis: chromatin compartments

FAN-C includes implementations of the most established analyses and measures for the characterisation of Hi-C matrix properties (Figs. 1c, 3a). Contact strength and the preference of contacts between certain genomic regions are particularly useful measures for gaining a global view of chromatin organisation. FAN-C implements several tools for this type of analysis, which are demonstrated using a high-resolution GM12878 dataset [25]:

i)   Contact distance decay plots: the average contact strength between loci separated by a certain distance, also called "expected contacts", is typically shown in a log-log plot of expected contacts vs distance (Fig. 3c). The slope and shape of the curve can inform about compaction of chromatin at various distance scales [1];

ii)  Observed/expected (O/E) transformation: a central transformation used by many analyses in which each pixel represents the (log2-)fold-change enrichment over the expected contact intensity for a region at that distance (Fig. 3d). Expected values are stored by FAN-C inside each matrix, allowing a fast, dynamic conversion of normalised into O/E contacts for various applications;

iii) Correlation matrices: the O/E matrix can further be transformed into a correlation matrix, in which each pixel i, j is calculated as the Pearson correlation coefficient between contacts in row i with column j (Fig. 3e, top). This highlights similarities and differences in contact profiles between loci and reveals the partitioning of regions into the so-called A and B compartments in a plaid-like pattern [1]. Computationally, these are assigned using the sign of the correlation matrices' first eigenvector (EV) (Fig. 3e, bottom). Due to the nature of EVs, positive entries do not necessarily correspond to the A and negative to the B compartment. FAN-C offers the option to integrate information from a genomic FASTA file, which

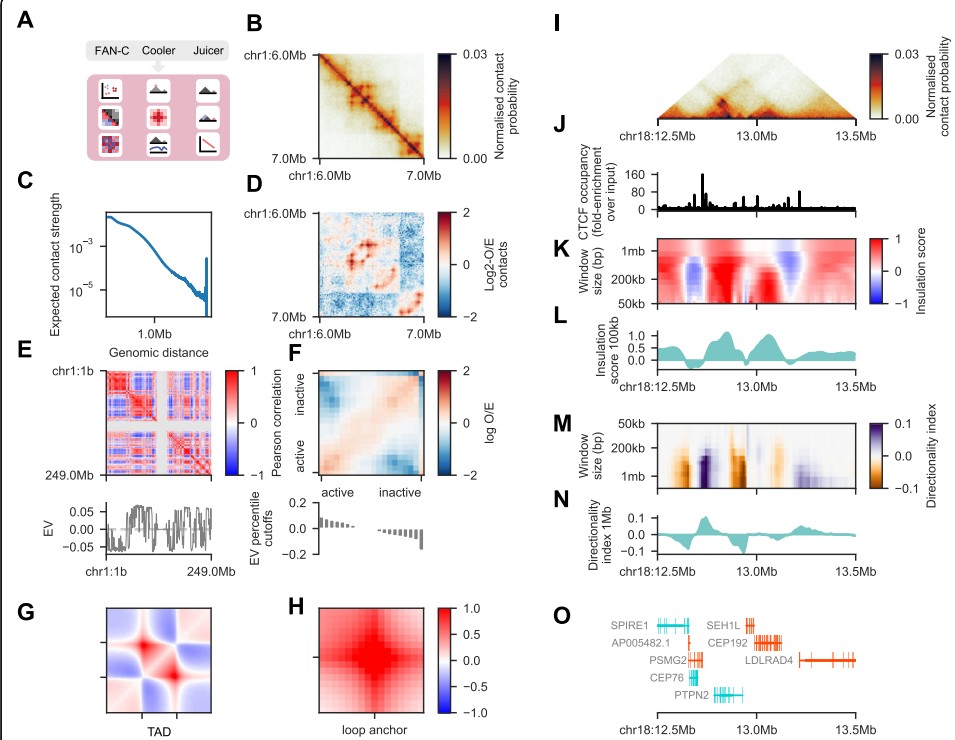

**Fig. 3** FAN-C analysis features. All analyses performed on GM12878 cells [25] on the 10 kb resolution matrix, unless otherwise noted. **a** Schematic representation of the analysis types available for FAN-C, Cooler, and Juicer matrices. **b** Hi-C matrix plot of a sample region with 10 kb resolution. **c** Log-log "Distance decay" plot of the expected normalised contact frequency against locus distance. **d** Log2-observed/expected (O/E) matrix for the same region as in **a**. **e** 500 kb resolution correlation matrix/A/B compartment plot of chromosome 1 (top) and its first eigenvector (EV) (bottom). **f** "Saddle plot" showing preferential interactions of active/active and inactive/inactive regions (top), and bar plot showing the cutoffs used for binning regions by the corresponding EV entry magnitude (bottom). Note the outlier on the far right. **g** Aggregate TAD plot showing the average log2-O/E in and around arrowhead domains [25]. **h** Aggregate loop plot showing the average log2-O/E at peaks called by HICCUPS [25]. **i–n** Example region on chromosome 18 highlighting additional analyses available in FAN-C and the possibility of "genome browser" style plotting. **i** Triangular Hi-C matrix plot. **j** Line plot showing CTCF occupancy (fold-change over input) as measured by ChIP. Data from GEO:GSM733752. **k** Heatmap showing insulation scores calculated using different window sizes. **l** Insulation score track for a window size of 100 kb. **m** Heatmap showing directionality index results for multiple window sizes. **n** Directionality index track for a window size of 1 Mb. **o** Gene plot using data from Gencode (v19) [57]

utilises the fact that the A compartment typically contains more GC-rich regions [1] to flip the EV entry signs accordingly. The magnitude of the EV entry corresponding to a region is a rough measure for the region's activity [1, 15];

iv) Saddle-plots: this helpful analysis allows the visualisation of interactions between A/B compartments of varying strength (Fig. 3f, top). To perform this analysis, regions are ordered and binned by their compartmentalisation strength (their entry in the correlation matrix EV) (Fig. 3f, bottom). The O/E values between regions of varying compartment strength provide a useful illustration of A and B segregation and can further be used to quantify the level of compartmentalisation in the whole genome [15] . A plot of cutoffs used for binning of regions is shown underneath the saddle plot (Fig. 3f, bottom). Unusually high or low EV entries, resulting, for

example, from noisy or low mappability regions, can cause artefacts in the saddle plot and are thus easily identifiable.

## Matrix analysis: TADs, chromatin loops, and aggregate analysis

High-resolution analyses of Hi-C matrices have revealed conserved matrix features that appear to be common across higher eukaryotes. These include topologically associating domains (TADs) [30–32], regions of increased self-interaction that are separated by insulating boundaries from neighbouring domains and are visible as squares in a Hi-C matrix (Fig. 3b), and chromatin loops [25], enriched discrete contacts between pairs of regions that show up as local areas of increased contact intensity in the matrix (Fig. 3b). FAN-C contains implementations of the most widely used algorithms for TAD and loop analysis:

i) Insulation score and directionality index: genomic regions between TADs, characterised by their strong insulating effect on neighbouring domains, can be identified using the insulation score [58] (Fig. 3k, l) or the directionality index [30, 59] (Fig. 3m, n). The resulting insulation tracks, quantifying the insulating effect of each region, can be exported to a range of established genomic formats, so they can easily be imported into genome browsers or used in other analysis pipelines.

ii) Chromatin loops: discrete peaks in the Hi-C matrix correspond to loops between genomic regions [25]. To identify these loops, FAN-C includes a CPU implementation of HICCUPS, a local-neighbourhood based loop calling algorithm [25], which can be parallelised on a computational cluster.

iii) Aggregate plots: to help with the identification of global trends across chromatin contact datasets, a genome-wide overview of the conformation around TADs, loops, or other genomic features such as promoters can be obtained with aggregate plots, which represent an average conformation around all regions of interest [15]. FAN-C implements the generation of aggregate plots from any list of regions or region pairs, with useful presets for TAD (Fig. 3g) and loop (Fig. 3h) aggregate plots. The aggregation process and the look of the aggregate matrix plot are highly customisable, for example by controlling size and resolution of the matrix, as well as colours and annotations of the final plot.

## Matrix comparison: highlighting and identifying differential features

A central task in Hi-C matrix analysis is the comparison of multiple datasets [41, 60–63]. FAN-C can systematically identify differences at all scales of the chromatin organisation hierarchy. Here, we use a previously published dataset of neuronal differentiation [18] to demonstrate the FAN-C steps used to find both global and local organisational distances in Hi-C data. Individual local changes can be visualised with FAN-C using targeted visualisation approaches.

First, using 'fanc compartments', we calculate the AB correlation matrix for embryonic stem cells (ESC), neuronal precursor cells (NPC), and cortical neurons (CN). The same command can then be used to generate AB compartment, or "saddle" plots, which show the enrichment in contacts between regions with different compartment eigenvector values (EV) relative to their expected values. Here, low EV values typically

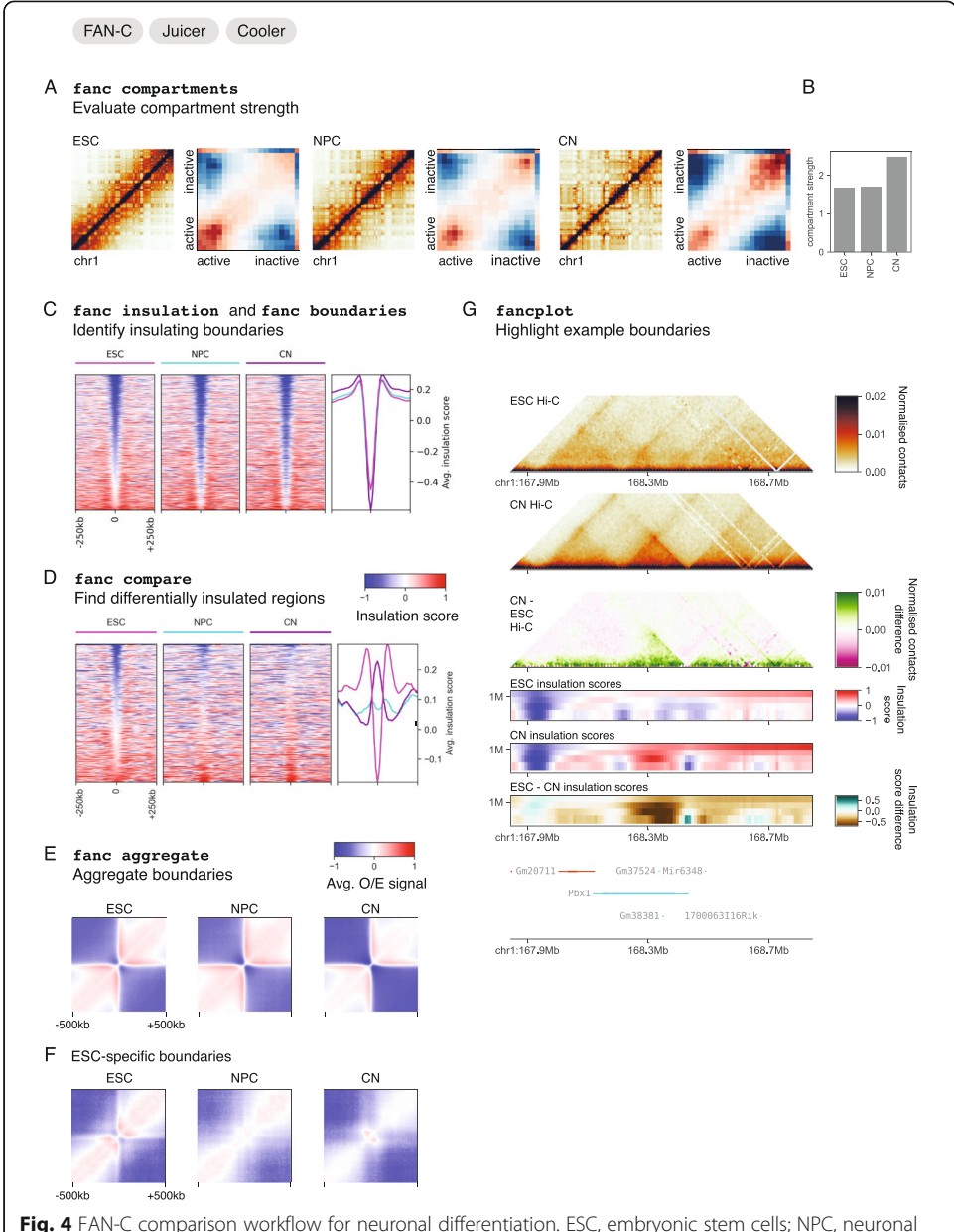

**Fig. 4** FAN-C comparison workflow for neuronal differentiation. ESC, embryonic stem cells; NPC, neuronal precursor cells; CN, cortical neurons. **a** Saddle plots showing contacts relative to expectation among regions with different compartment eigenvector values (binned by 2% percentiles). **b** Compartment strength barplot. **c** Heatmap of insulation scores at all boundaries in ESC, NPC, and CN, sorted by insulation score in CN. **d** Heatmap of insulation scores at differentially insulated regions between ESC and CN. **e** Aggregate matrices of 1mb windows centred at all boundaries in ESC, NPC, and CN. **f** Aggregate matrices of 1mb windows centred at ESC-specific boundaries. **g** Example of a differentially insulated region at the Pbx1 locus, showing Hi-C matrices for ESC and CN, a difference Hi-C matrix of CN- ESC, insulation scores of ESC and CN at various window sizes, insulation score difference between ESC and CN, and genes in the region, coloured by strand (orange = forward, cyan = reverse)

correspond to less transcriptionally active and high EV values to more active regions. From the saddle plots, it is directly evident that the compartment strength changes from ESC to CN, with particularly inactive regions gaining contacts in CN (Fig. 4a). This can be quantified with the compartment strength [15], showing an increase in compartmentalisation throughout ESC to CN differentiation (Fig. 4b).

Local differences in chromatin structure often involve changes in domain organisation, which reflect differences in insulation between neighbouring regions. We first calculate the insulation score [58] on the ESC, NPC, and CN samples at multiple window sizes using 'fanc insulation'. For a window size of 100 kb, we then find TAD boundaries in each sample by running 'fanc boundaries' with a cutoff of 1.0 for boundary strength—the difference in insulation score between the minimum at the boundary and the nearest maxima. Figure 4c shows a heatmap of the insulation score for the combined ESC, NPC, and CN boundaries, with the average insulation score for each sample plotted on the right.

To identify regions with local differences in insulation between the ESC and CN samples, we use 'fanc compare' to generate an insulation score difference track. Since this track can be directly used as input to 'fanc boundaries', we can directly identify regions where the insulation difference between ESC and CN is maximal. The heatmap in Fig. 4d shows the insulation score in all three samples only for differentially insulated regions between ESC and CN. An intuitive way to visualise (differential) boundaries is shown in Fig. 4e and f. Using 'fanc aggregate', we generate aggregate plots of 1mb size for all boundaries (Fig. 4e), and for differential boundaries (Fig. 4f).

Finally, we highlight one of the differentially insulated regions identified above using FAN-C plotting capabilities (Fig. 4g). We plot the normalised ESC and CN Hi-C matrices, as well as the difference between these two, which has been obtained using 'fanc compare'. Beneath the matrices, we show the insulation scores at multiple window sizes in each matrix, and the difference between insulation scores below. By including gene annotation in this locus, we can observe that the difference in insulation is caused by a TAD gain in CN at the Pbx1 gene, a transcription factor known to be important throughout neuronal development [64–67]. The visualisation of differences is independent of the approach used to detect them. Therefore, using FAN-C together with alternative approaches to detect differential contacts can also serve to examine chromatin conformation changes among samples.

### Additional comparison features

An additional option for global matrix comparisons is principal component analysis (PCA), which can provide a general overview of the similarity of several datasets by performing a pairwise comparison of matrix entries [14, 23]. FAN-C implements methods for PCA analysis of Hi-C contacts, such as for Hi-C library replicates and samples. Since not all pixels in a matrix are equally informative, e.g. regions far away from the diagonal or inter-chromosomal contacts are often dominated by noise, FAN-C includes a number of filters, such as distance between loci, or largest variance between samples, to only consider the most informative contacts in a matrix.

Finally, FAN-C comparison functions are not limited to objects created by FAN-C, but can be applied to any user-supplied in supported genomic formats (BED, GFF, Big-Wig, and more). The resulting tracks and matrices can be used as input to any FAN-C function in the same fashion as regular objects, including the visualisations outlined in the following section.

### Plotting: interactive and publication-ready visualisation of Hi-C and related data

FAN-C includes an implementation of an advanced yet easy to use plotting library for C-derived datasets (Fig. 1d). A number of diagnostic plots are generated as part of the fanc auto command, including filtering statistics for read pairs, biases in ligation frequency, and chromosomal coverage. Specific versions of the plots can also be produced individually, to allow for a thorough comparison of parameters used in an analysis. Plots related to Hi-C matrix-derived measures, such as correlation matrix, saddle, and aggregate plots (Fig. 3e–h) are part of the individual analysis functions. Plots for time-consuming analyses, such as aggregating matrices over a large number of regions, can easily be tweaked and adjusted without having to re-compute the entire analysis.

In addition to static plots, FAN-C also includes a basic genome browser that allows for the interactive exploration of Hi-C and additional genomic datasets. These include various different representations of Hi-C matrices: square (Fig. 3b); triangular (Fig. 3i); mirrored, in which two triangular Hi-C matrices are shown above and below a horizontal dividing line; and split, where the diagonal separates two different matrices in a square plot. A slice of a Hi-C matrix can also be visualised as a virtual 4C plot, which shows the strength of contacts between a specific genomic region and a genomic interval, as a line plot. This can be useful, for example, to visualise specific pairwise interactions, or even to detect genomic rearrangements such as translocations [23] or genome insertions [68]. All of the above matrix plots can also be used to display difference and fold-change maps (Fig. 4g).

Several plot types are available for region-based data in a standard genomic data format, including support for BED, GFF, BigWig, and Tabix-indexed files. These can be displayed as boxes coloured by strand, optionally grouped into layers by a user-defined attribute, or—in case they contain scores such as ChIP-seq tracks—as bar or line plots (Fig. 3j). Insulation score and directionality index results, which depend on a chosen window size parameter, have a dedicated plot type that visualises scores for multiple window sizes simultaneously in a heatmap (Figs. 3k, l, 4g), similar to the previously suggested "domainogram" [69]. Finally, genome annotations can be plotted with intron/ exon visualisations, as well as depicting strand information (Fig. 3o).

In addition to interactive visualisation, FAN-C includes a powerful plotting API for generating vector-based, publication-ready visualisations. Each type of interactive plot outlined above is also available through the API and is individually customisable. Since it is based on the major Python plotting library Matplotlib [70], it is easily extensible and can easily be integrated in existing plotting scripts. As a demonstration, all panels in Fig. 3 of this manuscript, apart from annotations and schematics, have been generated entirely using the FAN-C plotting API. This makes FAN-C not only useful for Hi-C matrix analysis, but also for users wanting to produce reproducible high-quality plots from pre-computed matrices to integrate alongside their existing visualisations.

### Conclusions

Here we introduce FAN-C as an open-source, versatile, flexible, and powerful tool for Hi-C analysis. FAN-C is bundled with an extensive documentation and instructions for obtaining and running FAN-C on sample datasets, available at https://fan-c.readthe-docs.io/. The FAN-C complete code is available at https://github.com/vaquerizaslab/fanc [71], and code to reproduce results from Figs. 3 and 4 is available at https://

github.com/vaquerizaslab/fanc-manuscript. The documentation includes detailed examples of how to use the command line tools and, for advanced applications, the versatile Python API. When designing FAN-C functionality, we have specifically tried to include the most widely-used measures and analyses with sensible defaults, while offering fine-grained control over analysis details. A side-by-side comparison with existing Hi-C analysis tools shows the broad spectrum of analysis options covered by FAN-C (Table 1). Due to its feature set and compatibility with the most established Hi-C formats, we envisage FAN-C to occupy a central position in many Hi-C pipelines.

## Methods

### Hi-C matrix generation

All Hi-C matrices presented in this work have been processed from raw FASTQ files using the 'fanc auto' command in FAN-C (0.9.0). BWA mem (0.7.17-r1188) was used for alignment. Data from human samples [25], was mapped to hg38 and data from mouse samples [18] to mm10. The following additional settings were used: *-q 3 --le-inward-cutoff 5000 --le-outward-cutoff 5000 --split-ligation-junction*.

### Hi-C matrix analysis

The code and commands for analyses shown in Figs. 3 and 4 of this work are deposited in the GitHub repository https://github.com/vaquerizaslab/fanc-manuscript. Pre-processed data for these analyses can be found in our FAN-C Keeper library https://keeper.mpdl.mpg.de/d/147906745b634c779ed3/. All commands were run with default settings and parameters unless noted otherwise.

### API-based analyses

The expected value ("distance decay") plot has been generated using the 'fanc.plotting.statistics.distance_decay_plot' function on 10 kb resolution matrices using only chromosome 1.

The AB compartment analysis has been performed using the 'fanc.ABCompartmentMatrix' class, using its 'from_hic' method to build the correlation matrix and its eigenvector method to calculate the correlation matrix eigenvector (excluding chromosome Y). The enrichment profile for the saddle plot has been generated using the 'enrichment_profile' method using percentiles 5, 10, …, 100, and was plotted using the 'fanc.plotting.statistics.saddle_plot' function.

TAD and loop aggregate plots have been generated using 'fanc.AggregateMatrix.from_regions', and 'fanc.AggregateMatrix.from_center_pairs', respectively, and plotted using the 'fanc.plotting.statistics.aggregate_plot' function.

Insulation and directionality index scores have been pre-computed using the 'fanc insulation' and 'fanc directionality' command line functions, respectively, with window sizes 2mb, 1mb, 750 kb, 500 kb, 400 kb, 300 kb, 200 kb, 150 kb, 100 kb, 90 kb, 80 kb, 70 kb, 60 kb, and 50 kb.

Region-based and matrix plots are generated entirely using the FAN-C plotting API in 'fanc.plotting'. Square matrix plots: 'SquareMatrixPlot'; Triangular matrix plots: 'TriangularMatrixPlot'; Line plots, such as CTCF occupancy, or individual insulation/

directionality tracks: 'LinePlot'; Plots with multiple scores ("flame plots"): 'GenomicVectorArrayPlot'; Genes: 'GenePlot'.

### Command-line based analyses

The AB compartment analysis was performed using the 'fanc compartments' command, running the command once to convert each Hi-C matrix (ESC, NPC, CN) [18] to an AB correlation matrix, a second time on the AB correlation matrices directly with the *--enrichment-profile* argument to produce the saddle plots with percentiles 5, 10, …, 100, and the mm10 genome FASTA for eigenvector orientation, and a third time with the *--compartment-strength* argument to calculate scores for compartmentalisation in each cell type.

Insulation scores have been calculated using the 'fanc insulation' command with the *--geom-mean* setting and the window sizes 50 kb, 75 kb, 100 kb, 250 kb, 500 kb, 1 mb, and 2 mb. TAD boundaries were called using 'fanc boundaries' on a window size of 100 kb with a minimum boundary score cutoff (*-s*) of 1.0. Insulation score difference tracks have been computed using 'fanc compare' with the *--comparison difference* argument. Equally, the difference matrix between CN and ESC has also been computed using 'fanc compare' with the *--comparison difference* argument.

Matrix and boundary plots have been produced using the 'fancplot' command, using *-p triangular*, and *-p square* for triangular, and square Hi-C matrices, respectively, *-p scores* for insulation score "flame" plots, and *-p gene* for plotting gene locations and orientations. Aggregate plots were produced using 'fanc aggregate' with a 1mb window. Insulation score heatmaps were made using Python 3.7 and the Matplotlib 'imshow' and 'plot' functions.

## Supplementary Information

---

**Additional file 1.** Review history.

---

### Acknowledgements
We thank Noelia Díaz, Benjamín Hernández-Rodríguez, Elizabeth Ing-Simmons, and Jahnavi Bhaskaran from the Vaquerizas laboratory for helpful discussions, suggestions for implementation and beta testing of FAN-C. We are grateful to Alexis G. Grimaldi for help with early implementations of code. We thank Tom Sexton (IGBMC, Strasbourg) for beta testing of the software. We are grateful to Elizabeth Ing-Simmons and Tom Sexton for providing comments on the manuscript. We also thank the authors of the following tools for feedback regarding the features and analysis options implemented in their software: Cooler, HiC-Pro, HiC-bench, HiCExplorer, HiCUP, HiCeekR, HicDat, Juicer, and TADbit. We thank the ENCODE Consortium and the Snyder laboratory (Stanford) for generating the CTCF dataset used in this study.

### Peer review information

### Review history
The review history is available as Additional file 1.

### Authors' contributions
Conceptualization, K. K, C.B.H. and J.M.V.; methodology, K.K. and C.B.H.; formal analysis, K.K. and C.B.H.; writing—original draft, K.K. and J.M.V.; writing—review and editing, K.K., C.B.H., and J.M.V.; funding acquisition, J.M.V.; supervision, J.M.V. The authors read and approved the final manuscript.

### Authors' information
Twitter handles: @kaukrise (Kai Kruse); @vaquerizasjm (Juan M. Vaquerizas).

## Funding

Work in the Vaquerizas lab is supported by the Max Planck Society, the Deutsche Forschungsgemeinschaft (DFG) Priority Programme SPP2202 'Spatial Genome Architecture in Development and Disease' [project number 422857230], the DFG Clinical Research Unit CRU326 'Male Germ Cells: from Genes to Function' [project number 329621271], the European Union's Horizon 2020 research and innovation programme under the Marie Skłodowska-Curie [grant agreement 643062 – ZENCODE-ITN], and the Medical Research Council, UK. Open Access funding enabled and organized by Projekt DEAL.

## Availability of data and materials

FAN-C is available at the Vaquerizas Laboratory GitHub page: https://github.com/vaquerizaslab/fanc [71]. The software is primarily written in Python 3 and is tested on macOS (10.14) and Linux (Scientific Linux 6.10) systems. It is licenced under the GNU General Public License version 3 (GPLv3). The version of FAN-C used in this manuscript (0.9.0), along with scripts to generate Figs. 3 and 4, are deposited in Zenodo [72].

Human Hi-C data used in Figs. 2 and 3 is available in the Gene Expression Omnibus (GEO) under the accession number GSE63525 [25]. Mouse Hi-C data used in Fig. 3 is also accessible in GEO under the accession number GSE96107 [18]. The CTCF ChIP-seq data used in Fig. 3 is available from ENCODE (https://www.encodeproject.org/) under the experiment accession ENCSR000DZN [73, 74]. The mm10 and hg38 genome files and annotations have been obtained from GENCODE [57].

FAN-C is built using the following Python packages: Biopython [75], Cooler [38], H5py (h5py.org), Matplotlib [70], Numpy [76], Pandas [77], Pillow (pillow.readthedocs.io), PyTables (pytables.org), PyYAML (pyyaml.org), Pybedtools [78, 79], Samtools and Pysam [80] (github.com/pysam/developers/pysam), Scikit-image [81], Scikit-learn [82], Scipy [83], Seaborn, gridmap (github.com/pygridtools/gridmap), intervaltree (github.com/chaimleib/intervaltree), msgpack (msgpack.org), progressbar2 (github.com/WoLpH/python-progressbar), and pyBigWig [84].

## Ethics approval and consent to participate

Not applicable.

## Consent for publication

Not applicable.

## Competing interests

The authors declare that they have no competing interests.

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

## 