## [**Additional file 1.** Review history. · Genome Biology]

Review History

First round of review

Reviewer 1

Were you able to assess all statistics in the manuscript, including the appropriateness of statistical tests used? There are no statistics in the manuscript.

Were you able to directly test the methods? Yes.

Comments to author:

Kruse et al. present here a new bioinformatics framework for Hi-C data analysis called FAN-C. Compared to existing solutions, FAN-C has the interest to gather in a single application many different steps of the analysis from raw data processing, to downstream analysis and visualization. The paper is clear and well written. Each part of the Hi-C analysis and FAN-C functions are well described.

The tool is well packaged, easy to install, and well documented.

Please, find below a few comments ;

Major comments :

- Table1

Please carefully check the Table1. Although I'm not an expert on all these tools, I already saw some mistakes.

For instance, HiC-Pro can export Juicer, Cooler and txt files, uses mapping quality for filtering, or filter re-ligation events (ligation errors ?)

Juicer can also export Txt file through the `bump` command

(<https://github.com/aidenlab/juicer/wiki/Data-Extraction>)

HiCexplorer also includes the `h5` format or allows Triangular Hi-C visualization.

- FAN-C seems to be able to work with input files from several existing tools such as Juicer, cooler, HiC-Pro, etc.

If working with .hic and .cool works well, Txt file import is not clear.

For instance, I made a test with an output file from HiC-Pro using :

`fanc compartments testdata_250000_iced.matrix` but the format was not recognized.

Could you please add an example for txt import in

<https://github.com/vaquerizaslab/fanc/tree/master/examples>

Minor comments

- Table 1 : 'Contaminant DNA' is specific to FAN-C. Please give additional details.

- Could you please give some details about the performances of FAN-C especially for the processing of high-resolution data ?

- Please, make FAN-C available through (bio)conda

Reviewer 2

Were you able to assess all statistics in the manuscript, including the appropriateness of statistical tests used? There are no statistics in the manuscript.

Were you able to directly test the methods? Yes.

Comments to author:

The manuscript "FAN-C: A Feature-rich Framework for the Analysis and Visualisation of Chromosome Conformation Capture Data" by Kruse et al. describes a novel bioinformatic resource to analyze chromosome conformation capture data, with particular focus on Hi-C experiments.

FAN-C features a flexible pipeline to generate interaction matrices from Hi-C sequencing reads by combining the most common alignment strategies (iterative mapping and restriction site splitting of chimeric reads). In addition, FAN-C includes a comprehensive collection of tools to analyze and visualize Hi-C experiments matrices, including the detection and characterization of compartments, TADs and loops. Importantly, these tools are available both as command line tools and as a Python API.

Although there are available pipelines that covered well the generation of interaction matrices from Hi-C sequencing reads (mentioned by the authors in Table1), FAN-C fills an important gap by simplifying downstream analysis and by providing command line tools that are designed to be used by a broader audience. Therefore, FAN-C has the potential to become a widely employed tool in the field of chromatin architecture and transcriptional regulation.

The current manuscript provides a comprehensive summary of the resources that are available through FAN-C. In addition, it serves as a guideline for the analysis that should be routinely performed upon generation of 3D chromatin datasets. Overall, I believe that the manuscript would be of great interest for the Genome Biology readership.

Nevertheless, there are certain issues that should be addressed by the authors:

1. The authors state that the alignment of the reads is performed with either Bowtie2 or BWA. However, it is not clear whether BWA alignment is performed in local mode (like in the distiller pipeline for instance <https://github.com/mirnylab/distiller-nf>) or the rescue of chimeric reads relies on restriction site splitting or iterative mapping instead.

2. I have experienced compatibility issues on cooler files (Abdennur and Mirny 2019) with many FAN-C functionalities. Specifically, I succeeded in loading cooler files and plotting some interaction matrices with the Python API, but could not perform other analyses like predicting compartments, TADs, loops or generating aggregate plots. The same tools worked flawlessly in my system when using hic files as inputs.

It might be possible that these issues relate to conflicts between the cooler (v=0.8.7) and FAN-C (v=0.8.27) versions that I am using. But, unfortunately, cooler files are not used as examples in the documentation, so it is difficult to spot where the problem is. This represents a major issue since, as stated by the authors, coolers are together with hic files (Durand et al. 2016) the two most popular storage formats for Hi-C interaction matrices.

It would be important to expand the documentation to the use of cooler files as inputs and to add notes on compatibility with cooler versions.

3. Most visualizations and analysis inside FAN-C are performed using normalized matrices. However, there are several methods for normalization and standard hic files are often loaded with KR, VC and VC_SQRT normalizations. I have not found any obvious way to choose between these normalization methods, or even to visualize raw interaction matrices. Importantly, there are Hi-C related protocols, such as HiChIP (Mumbach et al. 2016), where standard normalizations cannot be applied. Therefore, it would be useful if the authors enable the selection of different normalization options when using FAN-C.

4. I consider that the FAN-C Python API has the potential to become one the most attractive choices for many users performing Hi-C downstream analysis. But, while the documentation for the FAN-C command line suite is very precise and thorough, the API is not sufficiently documented in the current manuscript. Relevant analyses are not covered, such as the calculation of insulation scores and boundaries, the comparison of two interaction matrices, the comparison of insulation scores or the calculation of aggregate plots.

The authors should expand the documentation accordingly, in order to cover this lack of information.

5. Related with the previous comment, the authors state that all the panels of Figure 3 were generated using the FAN-C Python API, for which they provide a link to the code. However, many of those visualizations are not generated using dedicated functions available through the FAN-C API, but they require relatively complex array handling in Python instead.

It would be useful to include dedicated functions in the API to automatize some of the most recurrent plots (i.e. distance decay, saddle plots, boundary aggregates, tad aggregates and loop aggregates). Since aggregate plots, which are arguably the most challenging ones, already have dedicated command line tools, I believe that the effort to implement those features should be reasonable.

6. I found the analysis for differential insulation illustrated in Figure 4C and Figure 4D of great interest. As an additional suggestion, I believe that the visualization of insulation scores around predicted boundaries could be a nice feature to add in future releases of FAN-C.

Point-by-point response to reviewers' comments

FAN-C: A Feature-rich Framework for the Analysis and Visualisation of Chromosome Conformation Capture Data

We would first like to thank the reviewers for the overall positive evaluation of our work, and for the helpful and constructive comments. Following the reviewers' suggestions, we have now addressed all comments in this revised version of the manuscript. The main changes include:

- We have contacted the authors of all tools listed in Table 1 and asked them to review the feature set we have assigned to their respective tools. We have updated Table 1 with their suggestions accordingly.
- We have resolved issues around Cooler import and have added new documentation sections with detailed explanations of the handling of different input file types within FAN-C, both in the command line and the Python interface.
- We have processed the high-resolution GM12878 B-lymphocyte dataset from Rao et al. (2014) with 5 billion reads, and have added a manuscript section and Table with runtime information.
- We have implemented two new normalisation strategies, vanilla coverage and square root vanilla coverage, for use with protocols where matrix balancing cannot be applied or would be inappropriate.
- We have greatly expanded the Python API documentation, which now covers all of the central FAN-C functionality also present in the command line interface.

Below, we respond to each of the specific comments in detail (reviewer comments are in *blue italic* typography, our responses are in black)

Reviewer #1: Kruse et al. present here a new bioinformatics framework for Hi-C data analysis called FAN-C.

Compared to existing solutions, FAN-C has the interest to gather in a single application many different steps of the analysis from raw data processing, to downstream analysis and visualization.

The paper is clear and well written. Each part of the Hi-C analysis and FAN-C functions are well described.

The tool is well packaged, easy to install, and well documented.

We thank the reviewer for the positive assessment of our framework and manuscript.

Please, find below a few comments ;

Major comments :

- Table1

Please carefully check the Table1. Although I'm not an expert on all these tools, I already saw some mistakes.

For instance, HiC-Pro can export Juicer, Cooler and txt files, uses mapping quality for filtering, or filter re-ligation events (ligation errors ?)

Juicer can also export Txt file through the `bump` command (<https://github.com/aidenlab/juicer/wiki/Data-Extraction>)

HiCexplorer also includes the `h5` format or allows Triangular Hi-C visualization.

We have now contacted the authors of each tool listed in Table 1 and have updated the table accordingly. We heard back from all authors, except those of HiC-inspector, HOMER, and HiFive. For comprehensiveness, we are now also including more tools belonging to the same "ecosystems" in the comparison, such as pairtools and distiller in the Cooler, or TADkit for the TADbit ecosystem.

- FAN-C seems to be able to work with input files from several existing tools such as Juicer, cooler, HiC-Pro, etc.
If working with .hic and .cool works well, Txt file import is not clear.
For instance, I made a test with an output file from HiC-Pro using :
``fanc compartments testdata_250000_iced.matrix`` but the format was not recognized.

Could you please add an example for txt import in
<https://github.com/vaquerizaslab/fanc/tree/master/examples>

We apologise if this has been unclear before, and have now bundled a HiC-Pro example matrix with FAN-C (examples/hicpro). We have also added a section in the documentation that explains the TXT file import using this example:

https://vaquerizaslab.github.io/fanc/fanc-executable/fanc-generate-hic/fanc_helpers.html#fanc-from-txt-import-hic-from-text-file

and a more general tutorial on working with different file types:

<https://vaquerizaslab.github.io/fanc/fanc-executable/compatibility.html>

Minor comments

- Table 1 : 'Contaminant DNA' is specific to FAN-C. Please give additional details.

We thank the reviewer for pointing this out. We have added the following explanatory text (lines 107-109):

“More targeted, library-specific filters, such as a filter removing contaminating reads from a different organism, can be added as necessary.”

- Could you please give some details about the performances of FAN-C especially for the processing of high-resolution data ?

We have now included an additional paragraph (lines 130-139) with accompanying table (Table 2) that discusses FAN-C performance on the high-resolution GM12878 B-lymphocyte dataset from Rao and Huntley et al. (2014) with 5 billion reads. Since the amount of resources available to each user are bound to be different, we have normalised all runtimes to minutes per 100 million reads, as processed on a single thread. Running the calculations on multiple processes/threads will decrease runtime accordingly:

“While FAN-C was built for utility and compatibility, and not explicitly for performance, it is entirely possible to process and analyse high-resolution Hi-C datasets. Many FAN-C matrix generation steps can be parallelised, and the final cumulative processing time will depend on the resources the user is able to provide. As an example, we have processed the GM12878 human B-lymphocyte cell-line dataset (Rao et al. 2014) containing roughly 5 billion reads. Table 2 lists the approximate processing time on a Linux machine with an AMD Opteron Processor 6376, 2300 MHz, expressed as minutes per 100 million reads, normalised to the processing time on a single thread. Processing times for species with smaller genomes, such as *Drosophila melanogaster*, can be significantly lower. Once the final Hi-C matrix is obtained, FAN-C can retrieve up to 150 million contacts/pixels per minute from a FAN-C object.”

- Please, make FAN-C available through (bio)conda

We are currently in the process of making FAN-C available through bioconda. Our “genomic_regions” library, which FAN-C depends on, has already been approved by the bioconda team on GitHub. The FAN-C recipe is currently under review by the bioconda team. A change introduced into the recipe from the conda developers broke the continuous integration tests due to several dependency conflicts, and we are currently waiting for

this to be resolved. You can follow the progress here: <https://github.com/bioconda/bioconda-recipes/pull/23911>

If these conflicts cannot be resolved, it is likely that FAN-C will initially only be available for Python 3.6, provided the bioconda team agrees. This is largely out of our control, but we will continue to monitor the status of these packages and update the FAN-C recipe as soon as these issues have been resolved.

Reviewer #2: The manuscript "FAN-C: A Feature-rich Framework for the Analysis and Visualisation of Chromosome Conformation Capture Data" by Kruse et al. describes a novel bioinformatic resource to analyze chromosome conformation capture data, with particular focus on Hi-C experiments.

FAN-C features a flexible pipeline to generate interaction matrices from Hi-C sequencing reads by combining the most common alignment strategies (iterative mapping and restriction site splitting of chimeric reads). In addition, FAN-C includes a comprehensive collection of tools to analyze and visualize Hi-C experiments matrices, including the detection and characterization of compartments, TADs and loops. Importantly, these tools are available both as command line tools and as a Python API.

Although there are available pipelines that covered well the generation of interaction matrices from Hi-C sequencing reads (mentioned by the authors in Table1), FAN-C fills an important gap by simplifying downstream analysis and by providing command line tools that are designed to be used by a broader audience. Therefore, FAN-C has the potential to become a widely employed tool in the field of chromatin architecture and transcriptional regulation.

The current manuscript provides a comprehensive summary of the resources that are available through FAN-C. In addition, it serves as a guideline for the analysis that should be routinely performed upon generation of 3D chromatin datasets. Overall, I believe that the manuscript would be of great interest for the Genome Biology readership.

We thank the reviewer for the positive assessment of our work, and recognising the importance and potential impact on the chromatin community.

Nevertheless, there are certain issues that should be addressed by the authors:

1. The authors state that the alignment of the reads is performed with either Bowtie2 or BWA. However, it is not clear whether BWA alignment is performed in local mode (like in the distiller pipeline for instance <https://github.com/mirnylab/distiller-nf>) or the rescue of chimeric reads relies on restriction site splitting or iterative mapping instead.

We thank the reviewer for pointing this out. We are using BWA mem, which is a local alignment tool that natively supports chimeric alignments. It is also possible to enable restriction site splitting and/or iterative mapping on top of BWA's native chimeric read handling. We have amended the text accordingly (line):

“FASTQ files are mapped independently to a reference genome using either Bowtie2 or BWA mem - the choice of mapper is detected automatically from the genome index specified.”

2. I have experienced compatibility issues on cooler files (Abdennur and Mirny 2019) with many FAN-C functionalities. Specifically, I succeeded in loading cooler files and plotting some interaction matrices with the Python API, but could not perform other analyses like predicting compartments, TADs, loops or generating aggregate plots. The same tools worked flawlessly in my system when using hic files as inputs.

It might be possible that these issues relate to conflicts between the cooler (v=0.8.7) and FAN-C (v=0.8.27) versions that I am using. But, unfortunately, cooler files are not used as examples in the documentation, so it is difficult to spot where the problem is. This represents a major issue since, as stated by the authors, coolers are together with hic files (Durand et al. 2016) the two most popular storage formats for Hi-C interaction matrices.

It would be important to expand the documentation to the use of cooler files as inputs and to add notes on compatibility with cooler versions.

We thank the author for raising this issue, which we have also encountered independently after submission. We have implemented a number of fixes and improvements for Cooler files, and hope that these issues are now resolved. We have tested FAN-C with various Cooler files made available by the 4D Nucleome project and did not encounter any further problems. We are now also including FAN-C example data which has been converted to Cooler and Juicer format, so that tutorials in the documentation can be followed with the file format of personal preference. The results are highly consistent, apart from minor local variations due to the “zooming” and balancing processes applied separately by each tool.

We have also included new sections in the documentation explaining the work with different file formats in more detail:

<https://vaquerizaslab.github.io/fanc/fanc-executable/compatibility.html>
<https://vaquerizaslab.github.io/fanc/api/compatibility.html>

In case the reviewer encounters further complications with Cooler files, we would appreciate if an issue could be raised on the FAN-C GitHub page from an anonymous account, so that we can debug the problem in more detail.

3. Most visualizations and analysis inside FAN-C are performed using normalized matrices. However, there are several methods for normalization and standard hic files are often loaded with KR, VC and VC_SQRT normalizations. I have not found any obvious way to choose between these normalization methods, or even to visualize raw interaction matrices. Importantly, there are Hi-C related protocols, such as HiChIP (Mumbach et al. 2016), where standard normalizations cannot be applied. Therefore, it would be useful if the authors enable the selection of different normalization options when using FAN-C.

We thank the reviewer for the suggestion. We have addressed this shortcoming in two different ways:

- (i) We have added vanilla coverage and square-root vanilla coverage as normalisation options in FAN-C, in addition to the previously existing KR and ICE balancing options. These can be chosen with “—normalisation-method” in fanc auto and fanc hic on the command line, or via Hic.normalise in the API
- (ii) We have extended the Juicer loading syntax, so that Juicer Hi-C files can be loaded with the normalisation method of choice, either on the command line, or in the API via fanc.load. The syntax is /path/to/juicer/file.hic@<resolution>@<norm>.

It is already possible to visualise raw interaction matrices. On the command line, both square and triangular matrix plots support the "--uncorrected" argument, which is documented here:

https://vaquerizaslab.github.io/fanc/fancplot-executable/matrix_panel_types.html#triangular

In the Python API, TriangularMatrixPlot and SquareMatrixPlot similarly support plotting raw matrices via the "matrix_norm=False" parameter, as documented here:

<https://vaquerizaslab.github.io/fanc/api/plot/matrix.html#square-matrix>

4. I consider that the FAN-C Python API has the potential to become one the most attractive choices for many users performing Hi-C downstream analysis. But, while the documentation for the FAN-C command line suite is very precise and thorough, the API is not sufficiently documented in the current manuscript. Relevant analyses are not covered, such as the calculation of insulation scores and boundaries, the comparison of two interaction matrices, the comparison of insulation scores or the calculation of aggregate plots.

The authors should expand the documentation accordingly, in order to cover this lack of information.

We thank the author for expressing such great interest in the Python API of FAN-C. We have now added documentation in the areas specified by the reviewer, and expanded a few additional ones. We have also added a few more relevant example files to be used with the documentation.

<https://vaquerizaslab.github.io/fanc/api/api.html>

5. Related with the previous comment, the authors state that all the panels of Figure 3 were generated using the FAN-C Python API, for which they provide a link to the code. However, many of those visualizations are not generated using dedicated functions available through the FAN-C API, but they require relatively complex array handling in Python instead.

It would be useful to include dedicated functions in the API to automatize some of the most recurrent plots (i.e. distance decay, saddle plots, boundary aggregates, tad aggregates and loop aggregates). Since aggregate plots, which are arguably the most challenging ones, already have dedicated command line tools, I believe that the effort to implement those features should be reasonable.

We agree with the reviewer that the code generating specific panels of the figure in the manuscript could be greatly simplified. Hence, we have now added dedicated functions for distance decay, saddle, and all types of aggregate plots, accompanied by additional documentation in the relevant API sections.

Additionally, we have modified the Figure-3 code to make use of these new functions. There are still a couple of more complex code sections, however, these are necessary only for the specific figure layout in this case. We have added a comment in the code to reflect this.

6. I found the analysis for differential insulation illustrated in Figure 4C and Figure 4D of great interest. As an additional suggestion, I believe that the visualization of insulation scores

around predicted boundaries could be a nice feature to add in future releases of FAN-C.

We thank the reviewer for the comment. We also find the heatmap representation particularly useful, especially for insulation scores and their differences. We are currently working on a tool to create these kinds of heatmaps from any genomic track, supporting a wide range of file formats. However, we cannot provide an accurate estimate on a release date, as this is a work in progress. In the meantime, you can export insulation scores to BigWig using the command line or Python API as outlined in the documentation, and create similar heatmaps using a third-party tool such as deeptools.

Second round of review

Reviewer 1

The authors addressed all the points we previously raised. I do not have any additional comments.

Reviewer 2

The authors have successfully addressed all my concerns